# Acceptance and commitment therapy (ACT)-based programs for reducing symptoms of depression, anxiety and/or stress applied to the educational context: An exploratory review

**John Alexander Pedraza Palacios**👁️*

Faculty of Psychology, Universidad Santo Tomas, Bogotá, Colombia

👁 This sole author contributed to the creation of this work.
* japedrazap@gmail.com

## Abstract

This scoping review focused on characterizing Acceptance and Commitment Therapy (ACT)-based intervention programs to reduce symptoms of depression, anxiety, and/or stress in the educational context with youth and students. Twenty-six studies published between 2014 and 2023 were reviewed, originating mainly from the United States, Australia, Finland, and Canada. The programs implemented varied in format (websites, mobile applications, in-person or virtual sessions) and modality (group or individual). The results highlighted the effectiveness of ACT in improving well-being and facilitating students' academic completion. The ACT components addressed included acceptance, cognitive defusion, present moment, self as context, values, and committed action, organized into open, focused, and committed content. Additional benefits were identified such as social support in group formats and treatment personalization in individual sessions. However, more research is needed to optimize the implementation and dissemination of ACT in educational settings, ensuring its integration into student well-being programs.

## Introduction

Mental health is a complex phenomenon influenced by various social, environmental, biological and psychological factors [1]. In this sense, research has focused particularly on the university population, since numerous studies indicate that anxiety and depression disorders are more frequent among university students and young people compared to the general population [2]. The importance of analyzing mental health at this stage is due to the challenges inherent to youth development, such as the search for greater autonomy, the fulfillment of academic responsibilities, personal and family expectations, the development of sexuality, relationships and the construction of personal identity [3]. As a result, it is estimated that between 10% and 20% of young people experience or will experience symptoms associated with some mental

**Data availability statement:** All relevant data underlying the findings of this study are publicly available in [Zonedo] at [https://zenodo.org/records/18499315]. The dataset includes the search strategy, list of included studies, and data extraction matrix necessary to replicate the findings. The data supporting this study were obtained from publicly accessible bibliographic databases (Scopus, Web of Science, ProQuest One, Pubmed, Virtual Health Library (VHL) and Web of Science). The author did not have special access privileges. Interested researchers can access the data through the respective databases, according to their institutional subscriptions.

**Funding:** The authors received no specific funding for this work.

**Competing interests:** The authors have declared that no competing interests exist.

disorder, which has significant repercussions not only in the personal, academic, family and social spheres, but also in the economic context and in the health system [4].

Current approaches to treating the symptoms of the disorders include complementary or adjuvant interventions, such as psychological therapies focused on acceptance and mindfulness, known as third-generation therapies [5]. Among these, Acceptance and Commitment Therapy (ACT) stands out for its comprehensiveness and effectiveness [6]. The term ACT, which also means acting, reflects the essence of this approach, which encourages active and committed participation in life [7].

The Acceptance and Commitment Therapy (ACT) approach emphasizes the functional analysis of behavior within its context, being an effective option in the psychotherapeutic treatment of people with significant emotional problems [8]. According to the contextual philosophy that supports ACT, factors such as the environment, behavior, personal history and behavioral results are part of the context and must be considered during the therapeutic process. The main objective of ACT is to promote mindfulness, enhance psychological flexibility and decrease experiential avoidance [9] through six key processes: acceptance, cognitive defusion (open content), contact with the present moment, the self as a context (centered content), identification of values and committed actions (committed content) [10].

Research in this field suggests that therapeutic change is mainly explained by the development of psychological flexibility, which allows people to adapt to situational demands, maintain conscious contact with the present, and act in accordance with their personal values, even in the face of difficult internal experiences [11].

Programs based on Acceptance and Commitment Therapy (ACT) aimed at intervening in the symptomatology of emotional problems require research that analyzes their efficacy and effectiveness. To do so, it is essential that these studies include the randomization of participants, the use of standardized therapeutic manuals, the selection of clients with similar clinical profiles, the exclusion of populations with high comorbidity and, in general, the application of rigorous procedures to evaluate therapeutic change [12]. It should be noted that implementing and evaluating treatments in psychology is a complex process. For this reason, there is currently a large number of studies focused on effective and evidence-based interventions. These works seek to improve the health and well-being of young people and students, offering approaches and strategies that provide effective solutions to their difficulties, alleviating symptoms and considering the social and community context in which they operate [13].

In comparative studies between Acceptance and Commitment Therapy (ACT) and Cognitive Behavioral Therapy (CBT), two widely used psychological approaches for the treatment of various emotional and behavioral disorders [14], both interventions have been found to be effective in reducing symptoms, showing significant improvements in the short term. However, ACT tends to have more lasting effects on symptom reduction, so the choice between one intervention or the other will depend on the needs, preferences, and available resources of researchers or therapists [15].

This study aims to identify the characteristics of Acceptance and Commitment Therapy (ACT)-based programs used to reduce symptoms of depression, anxiety,

or stress in the educational context with students. The aim is to provide an updated overview of how these programs are being developed in schools and what results have been obtained in terms of reaching the population and their impact on mental health.

Although systematic reviews and meta-analyses have shown that ACT is implemented in various contexts and can be effective in reducing symptoms of anxiety, depression, and stress [16], there is still a need for evidence to understand its processes and mechanisms of change, including how its interventions are structured [17].

Acceptance and Commitment Therapy (ACT) protocols focus on addressing linguistic processes that are considered to be linked to both psychopathology and its treatment. When implementing an ACT strategy for a specific problem, interventions can be adapted to the characteristics and resources of the context and the target population, which gives rise to protocols that vary depending on the group they are aimed at. Likewise, even within the same protocol, differences may occur depending on the particularities of the population and the objectives established by the researchers [18].

## Method

### Search process and procedure

An exhaustive search was conducted in the following databases: Scopus, Web of Science, ProQuest One, Pubmed, Virtual Health Library (VHL), using the following search equation: (student OR university OR "higher education") AND (anxiety OR depression OR stress) AND "acceptance and commitment therapy" AND ("program based on acceptance and commitment therapy" OR "intervention program" OR "intervention based"). The equation was applied to the title, abstract, and keyword fields in each database. The review of titles, abstracts, and keywords was performed independently by a single researcher. In order to ensure the objectivity and reproducibility of the selection process, a transparent search strategy was established in advance, with explicit inclusion and exclusion criteria, which were systematically applied to all records, in accordance with PRISMA guidelines. The articles initially obtained were imported into Zotero software for collection, management, and review. Subsequently, studies that met the inclusion criteria were organized into a database available for consultation. Data synthesis was carried out using a narrative and descriptive approach, typical of scoping reviews. The review process was carried out from January 7, 2024, to June 20, 2024, during which time studies were added or excluded according to the previously defined criteria.

### Inclusion and exclusion criteria

The inclusion criteria considered: a) Time: last 10 years b) Languages: English, Portuguese, Spanish and French c) Focus: educational programs/interventions based on ACT d) Population: adolescents and/or young people, high school or university students e) Accessibility: Open access texts.

Exclusion criteria include: a) Unspecified structure: Programs/interventions that did not specify their structure in the article b) Insufficient basis in ACT: Programs/interventions that were not based on at least two of the three ACT columns (open, focused and engaged content) c). Non-relevant population: Unspecified population such as students, young people, adolescents, secondary school students, university students d). Restricted access: articles that were not freely available.

### Preliminary results

The initial search yielded 723 articles distributed in the following databases: Scopus (16 articles), Web of Science (51 articles), PubMed (83 articles), ProQuest One (567 articles) and Biblioteca Virtual en Salud - VHL (6 articles). A review of the titles, abstracts and keywords was carried out, which allowed us to identify and eliminate 566 articles that were not related to the objective of the study. Additionally, 55 duplicate articles and 76 articles that did not meet the inclusion criteria of the study in terms of population and programs/interventions based on Acceptance and Commitment Therapy (ACT) and free

access full texts were eliminated. Finally, 26 articles that met the established inclusion criteria were selected and coded for analysis (see Fig 1).

## Data extraction

Data were extracted into a matrix with the aim of collecting and structuring information from the studies included in this review. The matrix detailed a comprehensive description of each study, including: title, author(s), year, country, language, aim/purpose, population, type of study, and methodological design. Regarding program information, data were collected on the name of the program, the problem/disorder addressed, and outcome measures. Aspects related to the program structure included mode of delivery, number of sessions, duration, follow-up time, and reported outcomes/effectiveness. Finally, the components of Acceptance and Commitment Therapy (ACT) addressed were included, such as open-ended content (Acceptance/Cognitive Defusion), focused content (Present Moment/Self-Context), and engaged content (Values/Committed Action). This organization allowed for a systematic and detailed evaluation of each program and intervention, facilitating comparison and synthesis of findings in the review.

To comply with the parameters established by PRISMA [19], which ensure an exhaustive report of the key elements in this review, the following points were included in the writing and presentation of the document:

1. Title

2. Abstract

3. Introduction

4. Methods

5. Results

6. Discussion

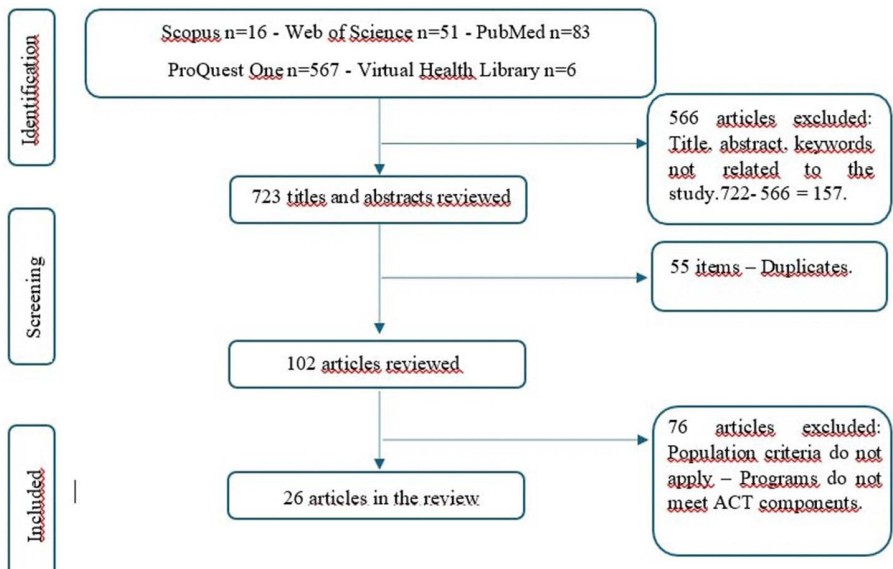

**Fig 1. Represents the process of identifying, screening, eligibility, and inclusion of the studies considered in the review.**

## Results

The objective of this review was to characterize the programs based on Acceptance and Commitment Therapy (ACT) used to reduce symptoms of depression, anxiety or stress in the educational context with young people and/or students. Twenty-six articles were reviewed to identify the characteristics implemented and evaluate the effectiveness of these protocols in addressing these problems. In addition, it was shown how the educational field adopts intervention programs aimed at improving or reducing symptoms in its students, which can positively influence the completion of their studies.

For the synthesis of results, statistical analyses performed and reported by the researchers of each study were taken into account. In the included studies, the effects of interventions were evaluated using statistical tests such as Student's t-test for related samples, analysis of covariance (ANCOVA), multivariate analysis of covariance (MANCOVA), and group comparison models, as appropriate for the design of each study.

The so-called levels of effectiveness were derived from the interpretation of the statistical results reported in the articles, considering both statistical significance and effect size when explicitly reported by the authors. In cases where effect sizes were not reported, the interpretation was based on the results of comparative analyses between groups or pre- and post-measurements described.

Since the review was conducted by a single researcher, no cross-verification of the extraction matrix was performed. However, previously defined selection, extraction, and synthesis criteria were applied systematically in order to promote the reproducibility of the process.

### Characteristics of included studies

The included studies were published between 2014 and 2023, with a higher prevalence of publications in the years 2016 (n = 4), 2022 (n = 6), and 2023 (n = 5). English was the predominant language of publication. Countries that excel in research and replication of Acceptance and Commitment Therapy (ACT)-based programs include the United States (n = 4), Australia (n = 5), Finland (n = 4), and Canada (n = 4). The aims of these studies varied, including the generation of digital mental health interventions to reduce the treatment gap, the adaptation of a health and wellness website for youth, and the evaluation of the effectiveness of a school-based mental health program. The populations targeted by the study, and also by the present review, were youth and high school and college students (see Table 1).

### Program information

Regarding the information collected to identify the characteristics of the programs, the following was highlighted: the names of the programs investigated through pilots or replications were DESTRESSIFY, ACTivate, STRONG MINDS, ABBT, MHEALTH, VRACT, KORSA, YOUTH COMPASS, LIVING TO THE FULLEST, PEER SUPPORT PROGRAM, DNA-V, YOLO, and HWBG (Health and Wellness for Girls). A prevalence was found in the programs KORSA (n = 3), COMPASS (n = 3), and YOLO (n = 2). It should be noted that some programs mentioned in this review do not have a stipulated name; in these cases, the articles refer to them as programs based on Acceptance and Commitment Therapy (ACT).

Specific problems and/or disorders addressed in the studies in this review include: psychological distress, anxiety, stress, depression, school stress, emotional crisis and psychological inflexibility. A higher prevalence of symptoms of depression, anxiety and stress was observed. Primary outcome measures are also of interest in assessing the decrease and improvement of these problems and/or disorders. The scales used include the Warwick-Edinburgh Mental Well-Being Scale (WEMWBS), the Depression, Anxiety and Stress Scale-Short Form (DASS-21), the Patient Health Questionnaire-9 (PHQ-9), the Social Interaction Anxiety Scale (SIAS), the short version of the Five Facet Mindfulness Questionnaire (FFMQ), the Multidimensional Psychological Flexibility (MPFI-24), the Psychological Well-Being Scale (PWB), the Multidimensional Experiential Avoidance Questionnaire (MEAQ), the General Health Questionnaire (GHQ-12), the CCAPS Scale, the Learning Behavior Comprehensive Rating Scale (LBCR), the Raynolds Adolescent Depression Scale-2

**Table 1.** Results and characteristics of the included studies.

| Author(s) | Country of origin | Objective |
|---|---|---|
| Becker, T.D.; Torous, J.B.;(2019). | USA | Bridging the treatment gap, given its potential for flexibility, cost-effectiveness and stigma reduction. |
| Brown, Menna; Lord, Emily; John, Ann. (2023). | UK. United Kimgdom | Adapting an existing health and wellbeing website for use by 16–24 year olds. |
| Rowan Burckhardt, Vijaya Manicavasagar, Philip J Batterham, Dusan Hadzi- Pavlovic. (2016). | Australia | A school-based mental health program combining positive psychology with acceptance and commitment therapy (Strong Minds) was developed. |
| Danitz SB; OrsilloSM (2014). | USA | To address some of these limitations, by developing and pretesting the efficacy of an ABBT intervention. |
| Emily Brenny Kroska Thomas; Gruichich, Tijana Sagorac; Maronge, Jacob M; Hoel, Sydney; Victory, Amanda; Stowe, Zachary N; Cochran, Amy. (2023). | USA | To assess the safety, feasibility, and efficacy of an ACT-based mHealth intervention. |
| Gorinelli, S.; Gallego, A.; Lappalainen, P.; Lappalainen, R.(2023). | Finland | To develop and investigate the effectiveness of an acceptance and commitment therapy (ACT)-based virtual reality intervention. |
| Grégoire S; Lachance L; Bouffard T; Dionne F. (2018). | Canada | To determine whether an Acceptance and Commitment Therapy (ACT)-based intervention was effective in improving psychological flexibility. |
| Grégoire, S; Beaulieu, F; Lachance, L; Bouffard, T; Vezeau, C; Perreault, M. (2022). | Canada | To assess the effect of an online peer support intervention based on Acceptance and Commitment Therapy using school and mental health indicators. |
| Grégoire, Simon; Lachance, Lise; Bouffard, Thérèse; Hontoy, Lysa-Marie; De Mondehare, Laurence. (2016). | Canada | To assess the effectiveness of an Acceptance and Commitment Therapy-based intervention. |
| Anna Guerrini Usubini, Roberto Cattivelli, Vanessa Bertuzzi, Giorgia Varallo, Alessandro Alberto Rossi, Clarissa Volpi, Michela Bottacchi, Sofia Tamini, Alessandra De Col, Giada Pietrabissa, Stefania Mannarini, Gianluca Castelnuovo, Enrico Molinari, Alessandro Sartorio. (2021). | Switzerland | To assess the effectiveness of a brief Acceptance and Commitment Therapy (ACT)-based intervention combined with treatment as usual (TAU). |
| Hämäläinen, Tetta, Kaipainen, Kirsikka, Keinonen, Katariina, Lappalainen, Päivi,Puolakanaho, Anne, Lappalainen, Raimo,Kiuru, Noona (2023). | Finland | To investigate the role of adherence and activity use in adolescents' gains. |
| Larsson, A.; Hartley, S.; McHugh, L. (2022). | Ireland | To assess the effectiveness of a brief web-based ACT intervention targeting a subset of ACT processes. |
| Levin ME; Krafft J; Hicks ET; Pierce B; Twohig MP.(2020). | USA | To compare the effects of a full online Acceptance and Commitment Therapy (ACT) intervention. |
| Liu, Y.; Chen, Y.; Liu, Z.; Zhang, Y.; Wu, M.; Zhu, Z. (2023). | China | To examine the efficacy of a Chinese version of the DNA-V model in a population of Chinese high school students. |
| Livheim, F; Hayes, L; Ghaderi, A; Magnusdottir, T; Hogfeldt, A; Rowse, J; Turner, S; Hayes, SC; Tengström, A. (2015). | Australia | To examine the effect of a brief intervention based on the principles of Acceptance and Commitment Therapy (ACT). |
| Morón, Juan José Macías; Valero- Aguayo, Luis. (2021). | Spain | To compare the efficacy of an Acceptance and Commitment Therapy (ACT)-based program versus a combined program adding strategies from Functional Analytic Psychotherapy (FAP) in first- and second-year high school students (n = 112). |
| Nedelcu, A; Grégoire, S. (2022). | Canada | To identify factors that facilitate or hinder the implementation of KORSA workshops. |
| Pots WT; Fledderus M; Meulenbeek PA; ten Klooster PM; Schreurs KM; Bohlmeijer ET. (2016). | Netherlands | To compare the efficacy of a web-based guided intervention based on Acceptance and Commitment Therapy (ACT) with an active control (expressive writing) and a wait-list control condition. |

*(Continued)*

**Table 1.** (Continued)

| Author(s) | Country of origin | Objective |
|---|---|---|
| Puolakanaho A; Lappalainen R; Lappalainen P; Muotka JS; Hirvonen R; Eklund KM; Ahonen TPS; Kiuru N. (2019). | Finland | To examine the effectiveness of a novel 5-week Finnish Internet- and mobile-delivered intervention program called Youth COMPASS. |
| Räsänen P; Lappalainen P; Muotka J; Tolvanen A; Lappalainen R. (2016). | Finland | To examine whether an online psychological intervention aimed at improving the well-being of university students could be an effective alternative. |
| Seyedeh, Maryam Noormohamadi; Arefi, Mokhtar; Afshaini, Karim; Kakabaraee, Keivan. (2022). | Iran | To examine the effect of Acceptance and Commitment Therapy (ACT) on the mental health of adolescents with emotional crisis (EB). |
| Van der Gucht, Katleen; Griffith, James W.; Hellemans, Romina; Bockstaele, Maarten; Pascal- Claes, Francis; Raes, Filip. (2017). | Belgium | To examine the effectiveness of an abbreviated, teacher-delivered, classroom-based Acceptance and Commitment Therapy (ACT) program. |
| Viskovich S; Pakenham KI. (2018). | Australia | To evaluate a 4-week web-based Acceptance and Commitment Therapy (ACT) mental health promotion program called YOLO. |
| Viskovich S; Pakenham KI. (2020). | Australia | To evaluate a 4-week web-based Acceptance and Commitment Therapy (ACT) mental health promotion intervention for college students. |
| White, Karen; Lubans, David R; Eather, Narelle. (2022). | Australia | To evaluate the feasibility and preliminary efficacy of a school-based health and wellness program (Health and Wellness for Girls: HWBG). |
| Zancan, Renata Klein; Constantinopolos, Laura Bohn; Pankowski, Bárbara Etchegaray; Bellini, Bárbara Diefenbach; Oliveira, Margareth da Silva. (2022). | Brazil | To explore undergraduate students' perceptions of their participation in an Acceptance and Commitment Therapy-based intervention. |

(RAAS-2), the Depressive Symptoms with the Dutch version of the CES-D and the Mental Health Continuous Form (MHC-SF).

## Program structure

Program structure included different delivery modalities, such as virtual, in-person, through mobile apps, websites, or bimodal that combined virtual and in-person. The number of sessions varied significantly between programs, with durations of 3, 4, 5, 6, 8, 9, 10, 12, and 16 sessions. This variability depended on the processes addressed and the time stipulated for participants to complete the program. The total duration also fluctuated depending on the program and the activities implemented for the target population, ranging from 2 to 12 weeks and up to 3 months. Generally, the duration was related to the number of sessions; for example, a 6-session program would likely last 6 weeks, with one session per week.

The application of the programs varied between group and individual approaches, and the follow-up time ranged from 3 weeks, 5 weeks, 1 month, 2 months, 3 months, 6 months, and 12 months. However, not all programs included a follow-up time, which represents a limitation when evaluating whether the results obtained initially are maintained over time. Finally, the results and degrees of effectiveness reported indicate that, in general, participants experienced significant improvements in several aspects of the primary measures, that is, the main results expected in the studies, as well as in the secondary measures, which are additional, although less critical, results. These measures were used to evaluate the improvements and the decrease in the problems addressed (see Table 2). However, one of the programs evaluated in this scoping review reports that the critical tests for ACT were not significant. No effect of the intervention was also found in a subgroup of students with mild to moderate depression symptoms.

**Table 2.** Program structure.

| Program | Delivery mode | Follow-up time | Results/degree of effectiveness |
|---|---|---|---|
| KORSA | In-person<br>Group<br>4 sessions<br>4 weeks | N/A | Students who participated scored highly in the program on all measures of psychological flexibility used in this study. |
| DESTRESSIFY | Mobile app<br>Individual<br>5 days a week for 4 weeks | N/A | Participants in the mindfulness app group were found to have significant improvements on measures of trait anxiety, general health, energy, and emotional well-being.bienestar emocional. |
| STRONG MINDS | In-person<br>Group<br>16 sessions<br>3 months<br>2 times a week | N/A | There was a statistically significant reduction in depression scores, stress scores, and DASS-Total scores for the Strong Minds condition compared to the control condition. |
| ABBT | In-person<br>Group<br>5 workshops<br>2 weeks | 3 months after the intervention | Students who participated in the workshop reported significantly lower depression and higher acceptance scores at 3-month follow-up. |
| MHEALTH | Mobile app<br>Individual<br>84 sessions available<br>6 weeks | 3 and 6 months | The findings indicated that the intervention was safe and feasible. The intervention increased values-based behavior, but did not decrease avoidance behavior. The intervention reduced depressive symptoms, but not perceived stress. |
| VRACT | In-person<br>Group<br>3 sessions<br>Each session of almost 2 hours | N/A | After attending all three VR sessions, participants in the intervention group recorded a significant decrease in social interaction and public speaking anxiety, fear of negative evaluation, and stress, and a significant increase in well-being, psychological flexibility, and self-compassion. |
| PEER SUPPORT PROGRAM | Virtual<br>Group<br>5 sessions<br>5 weeks | 5 weeks | Results indicate an improvement in psychological flexibility and a reduction in psychological rigidity after the intervention. |
| KORSA | In-person<br>Group<br>4 sessions<br>10 weeks | 1 month | Psychological flexibility indicators progressed. Students developed skills (ACT components). Symptoms associated with stress, anxiety, and depression decreased. Academic engagement increased. |
| ACT+TAU | In-person<br>Group<br>3 sessions<br>3 weeks | N/A | To be implemented. |
| ACT ONLINE | Website<br>Individual<br>12 sessions<br>6 weeks | N/A | ACT conditions produced equivalent, medium to large, improvements in the primary outcome of Mental Health Symptoms relative to the waitlist condition. Only the Engaged and Complete conditions improved the secondary outcome of Positive Mental Health relative to the waitlist condition. |
| DNA-V | Website<br>In-person<br>Group/Individual<br>6 sessions<br>6 weeks | 2 month | ACT shows more significant improvements in several aspects of mental health at follow-up. This may be because ACT can improve psychological flexibility. If people can accept, use defusion, and engage and act consistently with their values, they may experience fewer negative psychological events such as depression or anxiety, which will improve their mental health. |
| YOUTH COMPASS | Mobile Intervention<br>Individual<br>5 sessions<br>5 weeks | N/A | Statistically significant and expected changes were seen in general stress and academic optimism. Additional analyses also showed that those with higher stress in the initial phase of the study made greater positive gains in the interventions than those with low initial stress levels. |

*(Continued)*

| Program | Delivery mode | Follow-up time | Results/degree of effectiveness |
|---|---|---|---|
| YOUTH COMPASS | Website In-person 5 sessions 7 weeks | 12 month | Participants in the iACT intervention, which included two face-to-face sessions and weekly online contact, showed greater psychological, emotional, and social well-being and a reduction in stress symptoms. In addition, participants in the iACT group reported reductions in depression symptoms, greater life satisfaction, and higher self-esteem. The results were maintained at the 12-month follow-up. |
| YOLO | Website Individual 4 sessions 4 weeks | 12 weeks | Intervention participants improved from pre- to post-intervention on most primary aspects. The benefits of the intervention on most outcomes were maintained at follow-up despite a marginally significant decline in life satisfaction from post-intervention to follow-up, although it did not return to pre-intervention levels. |
| HEALTH AND WELL-BEING FOR GIRLS: HWBG | In-person Group 10 sessions 20 weeks | N/A | Participants in the HWBG programme reported high levels of satisfaction, most lessons (45 out of 50) were implemented as planned and adherence to the planned content of the HWBG programme was very high. Medium positive effects were observed on mental health (d = 0.45) and social health (d = 0.50). |
| ACTivate | Virtual Group 5 sessions 12 weeks | N/A | The use of participatory approaches to develop digital resources for health and wellbeing is recommended in established guidelines for digital health interventions. |
| BRIEF WEB-BASED ACCEPTANCE AND COMMITMENT THERAPY | Website Group 3 sessions 3 weeks | 3 weeks | The findings suggested that a brief web-based intervention focusing on these ACT processes has potential beneficial effects on general mental health and may therefore be useful in an approach to students' psychological health. |
| ACT EXPERIEN-TIAL ADOLESCENT GROUP. | In-person Group 8 sessions 8 weeks In-person Group 8 sessions 6 weeks | N/A | The Australian study demonstrated that on the primary outcome variable of depression (measured by RADS-2 total score), there was greater improvement for those in the ACT group format, with a large effect size. The Swedish study demonstrated that on the primary outcome of stress (measured by PSS), there was greater improvement for those in the ACT group format, with a large effect size. |
| ACT – FACT | In-person Group 3 sessions 4 weeks | N/A | The results indicate that both interventions (ACT and FACT) are effective in improving psychological well-being and life satisfaction, which were the main targets of the intervention. The ACT group is superior in psychological flexibility, mindfulness, and life satisfaction. |
| YOLO | Website Individual 4 sessions 4 weeks | N/A | YOLO participants improved post-intervention in depression, anxiety, stress, well-being, self-compassion, and life satisfaction. Lastly, alcohol and drug use did not change significantly. |
| KORSA | N/A | N/A | Facilitating factors: Management support, Colleague support, Efficiencies of promotion and hiring methods, Team encouragement. Main factors preventing this: Duration of workshops and Scheduling difficulties. |
| LIVING TO THE FULLEST | Website Individual 9 sessions in 3 parts 12 weeks | 6 to 12 months | The results show that in the short term, the ACT intervention was significantly more effective on the primary outcome measure and on most secondary out-come measures. ACT showed significantly greater reductions in depression. ACT was maintained at 12 months follow-up. |
| ACT | Individual 9 sessions 9 weeks | 1 month | Before the intervention, there was no difference between the love and rumination impact score between students in the experimental and control groups (p > 0.05); but after the intervention, the love and rumination impact score in the experimental group decreased (p < 0.05) |

*(Continued)*

**Table 2.** (Continued)

| Program | Delivery mode | Follow-up time | Results/degree of effectiveness |
|---|---|---|---|
| ACT | In person Group 4 sessions 120 minutes per week | N/A | For the 11 dependent variables, the critical tests for ACT were not significant. No effect of the intervention was also found in a subgroup of students with mild to moderate depression symptoms. |
| ACT | Presencial In person Group Individual 8 sessions 1 two-hour meeting for 2 months. | N/A | Regarding the usefulness of the program, some participants reported that the experience had a reach beyond university-related issues, such as coping with life in general and relationships with family members and others outside of campus. In addition, it was also possible to perceive in the statements the practice of self-care. |
| YOUTH COMPASS | Website/In person Individual 5 modules with approximately 90 exercises for 5 weeks | N/A | Adherent and committed users with relatively large intervention gains experienced a decrease in their stress, while less committed users without intervention experienced a substantial increase in their stress. |

## Acceptance and Commitment Therapy (ACT) components addressed

For this review, it was essential to recognize the components addressed in Acceptance and Commitment Therapy (ACT) in order to characterize the therapeutic processes that are being tested to reduce the symptoms of depression, anxiety, or stress. Therefore, the following contents were established based on the six ACT processes: acceptance or defusion (open content), present moment or self-context (centered content), and values or committed actions (committed content). Of the 26 articles reviewed and analyzed, it was found that 19 programs incorporate the six processes of the three contents, 2 programs only incorporated one process and one content, and 4 programs incorporated two processes and two contents.

Additionally, the programs that chose to include other elements in their training and execution were described, with the aim of improving the effectiveness in the treatment of the symptoms of the target population. Among these additional elements, programs that incorporated health promotion theories, health belief models, the theory of planned behavior, and the self-regulation model, along with a 12-week interactive intervention based on ACT, stood out. Another program was evaluated including some components of positive psychology. In addition, one program incorporated TAU focused on physical rehabilitation, while another included expressive writing according to the PennBaker method. Other programs integrated physical activities such as yoga, pilates, fun games, and competitions. Finally, one ACT program was compared with elements of Functional Analytic Acceptance and Commitment Therapy (FACT) (see Table 3).

## Discussion

Considering that the objective of this review was to characterize intervention programs based on Acceptance and Commitment Therapy (ACT) to reduce symptoms of depression, anxiety or stress in the educational context with young people and students, the results obtained showed a series of programs that have been tested in this area. These programs not only seek to improve the symptoms associated with these disorders, but also to promote the well-being of students and facilitate the successful completion of their academic process. 26 studies that met the inclusion criteria were reviewed to identify the key characteristics that configure these programs.

**Table 3. ACT programs and components addressed.**

| Program | Inclusion of open content (Acceptance or defusion) | Inclusion of focused content (Present moment or self-context) | Inclusion of committed content (Committed values or actions). | Inclusion of other elements |
|---|---|---|---|---|
| ACTivate | N/A | Present moment | N/A | Health Belief Models, Theory of Planned Behavior; Plan, Do, Study, Act, and Self-Regulation Model. |
| STRONG MINDS | Defusion. Acceptance. | Present moment I context | Values Committed actions | Positive Psychology. |
| ABBT | Aceptance | N/A | Values | N/A |
| MHEALTH | Aceptance. Defusion | Present moment | Values | N/A |
| VRACT | Aceptance. Defusion | Present moment I context | Values Committed actions | N/A |
| KORSA | Aceptance. Defusion | Present moment I context | Values Committed actions | N/A |
| PEER SUPPORT PROGRAMS | Aceptance. Defusion | N/A | Values Committed actions | N/A |
| ACT+TAU | Aceptance. Defusion | Present moment I context | Values Committed actions. | TAU- Physical Rehabilitation. |
| YOUTH COMPASS | Aceptance. Defusion | Present moment I context | Values Committed actions | N/A |
| BRIEF WEB-BASED ACCEPTANCE AND COMMITMENT THERAPY | Defusion | Present moment I context | N/A | N/A |
| ACT ONLINE | Aceptance. Defusion | Present moment I context | Values Committed actions | N/A |
| DNA-V | Defusion | I context | Values Committed actions | N/A |
| ACT EXPERIENTIAL ADOLESCENT GROUP | Aceptance. Defusion Aceptance Defusion | Present moment I context Present moment I context | Values Committed actions Values Committed actions | N/A TAU School Nurse. Individual counseling for 2 and 8 sessions. |
| ACT. - FACT | Aceptance. Defusion | Present moment I context | Values Committed actions | Comparison to a FACT program |
| LIVING TO THE FULLEST | Aceptance. Defusion | Present moment I context | Values Committed actions | PennBaker Expressive Writing |
| ACT | Aceptance. Defusion | I context | Values Committed actions | N/A |
| YOLO | Aceptance. Defusion | Present moment I context | Values Committed actions | N/A |
| HEALTH AND WELL-BEING FOR GIRLS: HWBG | Aceptance. | Present moment | Values | Physical Activity – Yoga – Pilates- Fun and Competitive Games. |
| ACT | Aceptance. Defusion | Present moment | N/A | N/A |
| DESTRESSIFY | Aceptance | Present moment | Values | N/A |

It has been observed that ACT-based programs are delivered in various ways, including websites, mobile applications, face-to-face or virtual sessions, both in group and individual formats, depending on implementation preferences. In a meta-analysis review conducted by Gloster [20], the implementation of ACT in group formats was highlighted, highlighting additional benefits such as social support among participants and the opportunity to practice therapeutic skills in an

interactive and collaborative context. On the other hand, Ponce-Alencastro [21] presented a case study on the treatment of social anxiety through individual sessions, managing to reduce anxiety levels and increase psychological flexibility, which underlines the importance of ACT in individualized clinical practice. A study conducted by Levin [22], based on the web for mental health problems in university students, demonstrated significant improvements in general distress, social anxiety, depression, academic concerns and positive mental health, validating the website as an effective alternative for the care of psychological problems. These findings provide evidence in this review on the characteristics of ACT-based programs that allow their adaptation to multiple delivery formats, being crucial for their effectiveness in diverse contexts and populations, thus reflecting a variety of modalities in which the therapy can be implemented.

Similarly, programs specifically designed to treat symptoms of depression, anxiety, and stress have been observed, suggesting that the use of a single program may reduce the symptomatology of these disorders. This is because ACT has been shown to be effective in improving individuals' psychological flexibility and reducing experiential avoidance. A meta-analysis of 39 randomized controlled trials by A-Tjak [23] on the efficacy of Acceptance and Commitment Therapy (ACT) for physical and mental health problems concluded that ACT is effective in treating anxiety disorders, depression, addiction, and somatic health problems, comparing favorably with other established psychological interventions. Hayes [24] also note that ACT helps people respond flexibly and effectively to the changing demands of life, which contributes to reducing the emotional rigidity associated with symptoms of depression, anxiety, and stress.

Among the components of ACT addressed in the programs, 19 of the 26 included studies address all components of the therapy with the goal of increasing psychological flexibility. Acceptance and Commitment Therapy (ACT) is distinguished by its holistic approach that integrates six fundamental processes: acceptance, cognitive defusion, contact with the present, observing self, values clarification, and committed action. According to Hayes and Pierson [25], the integration of these six processes in a therapeutic treatment helps both therapists and patients understand how each contributes to psychological well-being and how they interact to promote positive change.

Empirical research supports the effectiveness of ACT, showing that the application of these six processes is associated with significant improvements in a wide range of mental disorders [26]. Furthermore, the effectiveness of the programs requires the applicability of ACT in diverse contexts, including educational settings. The flexibility and adaptability of ACT allow its integration into intervention programs tailored to the specific needs of different populations and situations [27].

Among the results reported in the reviewed articles, effect sizes ranging from moderate to large stand out. These findings indicate that ACT-based interventions not only generate statistically significant changes, which could suggest an appreciable impact on symptom reduction, but also highlight the effectiveness of programs in educational contexts (Burckhardt et al. 2016; Brown et al., 2023).

However, it is important to note the existence of barriers that may hinder the implementation of these interventions in educational settings. Their application may require significant resources for training, supervision, and adaptation of materials, which represents a particular challenge in systems with limited resources [28]. Furthermore, obstacles may arise related to a lack of knowledge of ACT principles and a preference for traditional methods, factors that can generate resistance to its adoption [29].

For Petersen [30] more research is needed to better understand how to implement and disseminate ACT among young people. However, current evidence shows the effectiveness of ACT models in various settings, including educational settings. Given th urgent need to address mental health issues in young people, ACT is presented as an evidence-based and highly promising therapy to help this population. The high evidence and the possibility of future research that continues to demonstrate its effectiveness in this population and in the educational context are related to ACT's goal of increasing psychological flexibility [31]. This flexibility is linked to students' engagement and progression in their studies, as well as improving their well-being. Therefore, it is essential to continue piloting and replicating the programs until they become an integral part of student well-being processes.

Among the limitations of this study, it should be noted that several of the programs do not include long-term follow-up, which limits the assessment of the sustainability of the effects of the ACT intervention over time. The review was limited to freely accessible texts, which may have excluded relevant studies that could provide further information on the effectiveness of ACT programs in educational contexts. Having only one reviewer of the articles reduces the possibility of a more comprehensive assessment that considers different points of view and enriches the analysis of the findings. Finally, no formal assessment of the methodological quality or risk of bias of the included studies was performed, as this procedure is not a mandatory requirement in scoping reviews. Therefore, the results should be interpreted as a description of the current state of the evidence and not as a definitive assessment of the effectiveness of the interventions analyzed.

## Conclusions

This scoping review study has allowed us to characterize the programs based on Acceptance and Commitment Therapy (ACT) aimed at reducing symptoms of depression, anxiety and/or stress in young people and students in educational contexts. The results show that these programs have a structure and ACT components addressed that showed results not only in improving the mental health of students, but also promoting their general well-being and facilitating the successful completion of their studies. This suggests that ACT is a promising therapy to be implemented in educational settings. In addition, it highlights the importance of continuing research and piloting of these programs to effectively integrate them into student well-being processes.

## Supporting information

**S1 Data. Contains the matrix used for extracting information from the included studies, detailing relevant variables such as authors, year of publication, sample characteristics, methodology, and main results.**
(XLSX)

**S1 File. PRISMA checklist: Includes the completed PRISMA checklist, which demonstrates compliance with the methodological criteria for conducting systematic reviews.**
(DOCX)

## Author contributions

**Investigation:** John Alexander Pedraza Palacios.

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
