## [Decision Letter · Decision Letter 0]

11 Mar 2025

PMEN-D-24-00459

Acceptance and commitment therapy (ACT)-based programs for reducing symptoms of depression, anxiety and/or stress applied to the educational context: a scoping review.

PLOS Mental Health

Dear Dr. Pedraza Palacios,

Thank you for submitting your manuscript to PLOS Mental Health. After careful consideration, we feel that it has merit but does not fully meet PLOS Mental Health’s publication criteria as it currently stands. Therefore, we invite you to submit a revised version of the manuscript that addresses the points raised during the review process.

We look forward to receiving your revised manuscript.

Kind regards,

Bochra Nourhene Saguem, M.D.

Academic Editor

PLOS Mental Health

Journal Requirements:

1. We noticed you have some minor occurrence of overlapping text with the following previous publication(s), which needs to be addressed:

- https://contextualscience.org/fr/book/export/html/226

- DOI:10.1016/j.jad.2019.11.154

- https://doi.org/10.1515/ijamh-2019-0096

In your revision ensure you cite all your sources (including your own works), and quote or rephrase any duplicated text outside the methods section. Further consideration is dependent on these concerns being addressed.

2.  We ask that a manuscript source file is provided at Revision. Please upload your manuscript file as a .doc, .docx, .rtf or .tex.

Additional Editor Comments (if provided):

Reviewers' comments:

Reviewer's Responses to Questions

**Comments to the Author**

1. Does this manuscript meet PLOS Mental Health’s publication criteria? Is the manuscript technically sound, and do the data support the conclusions? The manuscript must describe methodologically and ethically rigorous research with conclusions that are appropriately drawn based on the data presented.? Is the manuscript technically sound, and do the data support the conclusions? The manuscript must describe methodologically and ethically rigorous research with conclusions that are appropriately drawn based on the data presented.

Reviewer #1: Partly

Reviewer #2: Partly

2. Has the statistical analysis been performed appropriately and rigorously?

Reviewer #1: N/A

Reviewer #2: N/A

3. Have the authors made all data underlying the findings in their manuscript fully available (please refer to the Data Availability Statement at the start of the manuscript PDF file)?

The PLOS Data policy requires authors to make all data underlying the findings described in their manuscript fully available without restriction, with rare exception. The data should be provided as part of the manuscript or its supporting information, or deposited to a public repository. For example, in addition to summary statistics, the data points behind means, medians and variance measures should be available. If there are restrictions on publicly sharing data—e.g. participant privacy or use of data from a third party—those must be specified.requires authors to make all data underlying the findings described in their manuscript fully available without restriction, with rare exception. The data should be provided as part of the manuscript or its supporting information, or deposited to a public repository. For example, in addition to summary statistics, the data points behind means, medians and variance measures should be available. If there are restrictions on publicly sharing data—e.g. participant privacy or use of data from a third party—those must be specified.

Reviewer #1: No

Reviewer #2: Yes

4. Is the manuscript presented in an intelligible fashion and written in standard English?

Reviewer #1: Yes

Reviewer #2: Yes

5. Review Comments to the Author

Reviewer #1: Thank you for giving me the opportunity to review your manuscript PMEN-D-24-00459, entitled 'Acceptance and commitment therapy (ACT)-based programs for reducing symptoms of depression, anxiety and/or stress applied to the educational context: a scoping review" for PLOS Mental Health.

While this study contributes to the understanding of the role of ACT, I recommend major revisions before it can be considered for publication. As currently presented, the manuscript does not adequately explain the importance of reviewing ACT. Below some points for revision:

1. Purpose of the Study:

You have made a good effort to bring together background information. Nevertheless, the manuscript lacks a clear articulation of its purpose, which is essential for establishing the rationale behind this scoping review. Additionally, it would be important to highlight what distinguishes this study from previous review studies. I recommend that the author address these critical aspects in the introduction and give a larger overview of what has already been done.

2. Literature Search and Timeframe:

The ending date of the literature search is not specified. Please provide these dates to ensure the study's transparency and reproducibility.

3. Data Extraction and Analysis:

The methodology section should include detailed information about the data extraction process, specifically how reviewers collaborated and ensured inter-rater reliability. How have the results been double-checked? What is the quality of the included studies? If this has not been done, it would be important to mention it clearly in the limitations.

4. Exclusion Criteria:

It is unclear what criteria were used to exclude a substantial number of studies during the screening stage. To justify this decision, a detailed explanation of the exclusion standards is required. I would advise making a table with inclusion and exclusion criteria.

5. Main Findings/Results:

I appreciate the authors’ effort in summarizing and reviewing the findings; the presentation seems a bit broad and scattered. To improve readability and coherence, I suggest summarizing the key findings of the review in tables organized in correspondence with the study's aims. I would recommend making the tables more compact, tailored to the study objectives, and also including the p-value of the included studies (for example, if you write, that’s highly effective). Adapt the table to the APA style. Please add information about the study design of the included studies.

6. Discussion: Refer to the studies found and discuss the results found and then put it into a larger context/actual knowledge. Also, you are talking about very good evidence; how can you highlight that? Additionally, I suggest to include the limitations in the discussion section.

7. References:

To facilitate comparison, I would propose to mark the references included in the scoping review with an asterisk (*) to distinguish them from other cited articles.

Reviewer #2: Strengths of the Study

1. The study is highly relevant, as mental health concerns among students are increasing, and Acceptance and Commitment Therapy (ACT) has gained recognition as an effective intervention.

2. The manuscript systematically reviews ACT-based interventions from 2014 to 2023, covering multiple formats (e.g., in-person, mobile apps, online interventions) and various educational settings across different countries.

3. The inclusion and exclusion criteria are well-defined, and the PRISMA guidelines ensure transparency in the review process.

4. The classification of ACT processes into open, focused, and engaged components provides clarity and aligns well with the theoretical framework of ACT.

5. The study captures diverse implementation strategies, including web-based, mobile, and group interventions, making the findings applicable across various educational settings.

6. The discussion on how group-based ACT programs provide social support is valuable for future implementation and program design.

7. The study acknowledges the need for more rigorous randomized controlled trials (RCTs) and long-term follow-ups, emphasizing areas that require further investigation.

Weakness of the study

1. While the study summarizes 26 research articles, it lacks a critical evaluation of their methodological quality. It would be beneficial to assess: The risk of bias in included studies, Differences in study designs and their impact on findings and Limitations in sample size or generalizability of individual studies.

2. The effectiveness of ACT is mentioned, but there is no comparative analysis of ACT against other interventions (e.g., cognitive-behavioral therapy, mindfulness-based interventions).

3. Quantitative effect sizes from the reviewed studies would strengthen claims about ACT’s efficacy.

4. The paper touches on the need for more research on implementation but does not extensively discuss barriers such as: Cost and feasibility in different educational settings, Potential resistance from students, educators, or policymakers and Differences in ACT training quality among facilitators.

5. The study provides a strong foundation in ACT theory but could further integrate frameworks such as: Psychological flexibility as a mediating mechanism and Behavior change models explaining how ACT interventions work in educational settings.

6. It could also discuss potential dropout rates and adherence challenges in ACT-based interventions.

7. A summary table comparing ACT effectiveness in different modalities (e.g., online vs. in-person) with key outcome metrics would be helpful.

8. Given the focus on educational settings, more actionable recommendations for educators, policymakers, or mental health practitioners would enhance the manuscript’s applicability.

6. PLOS authors have the option to publish the peer review history of their article (what does this mean?). If published, this will include your full peer review and any attached files.). If published, this will include your full peer review and any attached files.

**Do you want your identity to be public for this peer review?** For information about this choice, including consent withdrawal, please see our Privacy Policy..

Reviewer #1: No

Reviewer #2: No

---

## [Decision Letter · Decision Letter 1]

23 Nov 2025

PMEN-D-24-00459R1

Acceptance and commitment therapy (ACT)-based programs for reducing symptoms of depression, anxiety and/or stress applied to the educational context: a scoping review.

PLOS Mental Health

Dear Dr. Palacios,

Thank you for submitting your manuscript to PLOS Mental Health. After careful consideration, we feel that it has merit but does not fully meet PLOS Mental Health’s publication criteria as it currently stands. Therefore, we invite you to submit a revised version of the manuscript that addresses the points raised during the review process.

We look forward to receiving your revised manuscript.

Kind regards,

Bochra Nourhene Saguem, M.D.

Academic Editor

PLOS Mental Health

Journal Requirements:

Reviewers' comments:

Reviewer's Responses to Questions

**Comments to the Author**

1. If the authors have adequately addressed your comments raised in a previous round of review and you feel that this manuscript is now acceptable for publication, you may indicate that here to bypass the “Comments to the Author” section, enter your conflict of interest statement in the “Confidential to Editor” section, and submit your "Accept" recommendation.

Reviewer #1: (No Response)

Reviewer #3: (No Response)

2. Does this manuscript meet PLOS Mental Health’s publication criteria? Is the manuscript technically sound, and do the data support the conclusions? The manuscript must describe methodologically and ethically rigorous research with conclusions that are appropriately drawn based on the data presented.? Is the manuscript technically sound, and do the data support the conclusions? The manuscript must describe methodologically and ethically rigorous research with conclusions that are appropriately drawn based on the data presented.

Reviewer #1: Yes

Reviewer #3: Yes

3. Has the statistical analysis been performed appropriately and rigorously?

Reviewer #1: N/A

Reviewer #3: N/A

4. Have the authors made all data underlying the findings in their manuscript fully available (please refer to the Data Availability Statement at the start of the manuscript PDF file)?

The PLOS Data policy requires authors to make all data underlying the findings described in their manuscript fully available without restriction, with rare exception. The data should be provided as part of the manuscript or its supporting information, or deposited to a public repository. For example, in addition to summary statistics, the data points behind means, medians and variance measures should be available. If there are restrictions on publicly sharing data—e.g. participant privacy or use of data from a third party—those must be specified.requires authors to make all data underlying the findings described in their manuscript fully available without restriction, with rare exception. The data should be provided as part of the manuscript or its supporting information, or deposited to a public repository. For example, in addition to summary statistics, the data points behind means, medians and variance measures should be available. If there are restrictions on publicly sharing data—e.g. participant privacy or use of data from a third party—those must be specified.

Reviewer #1: Yes

Reviewer #3: Yes

5. Is the manuscript presented in an intelligible fashion and written in standard English?

Reviewer #1: Yes

Reviewer #3: Yes

6. Review Comments to the Author

Reviewer #1: Comments to Author:

Thank you for the careful revision. I have a few small clean-ups before signing off:

1. For clarity and transparency, please indicate the exact date of the database search (rather than only a two-month period) and note that this refers to the stage before exclusions. This could be added on page 5.

2. As is customary in scientific reporting, the limitations should be presented within the discussion section rather than placed afterwards. This can be effectively achieved by adding a 'Strengths and Limitations' subsection at the end of the discussion.

Reviewer #3: 1. The description of the literature search period is unclear (“The review was carried out between the month of May, until June 20, 2024”). Please specify the exact start date to enhance reproducibility.

2. The manuscript states that “a review of the titles, abstracts, and keywords was carried out”, but as this was a single-author review, it is unclear how objectivity and reproducibility of the screening process were ensured. Although this limitation was acknowledged in the response to reviewers, the Methods section should explicitly state that the screening was conducted independently by one researcher.

3. While the manuscript notes the use of a data extraction matrix, it does not sufficiently describe how the data were synthesized. Please clarify how “levels of effectiveness” were derived across studies. The statement that “effect sizes ranging from moderate to large ” should be supported with a description of how these effects were evaluated. In addition, indicate whether the data extraction matrix was completed by a single researcher or if any form of cross-checking was performed.

4. It would also be helpful to state in the Methods section whether a quality assessment of included studies was performed. If not, this should be explicitly acknowledged as a limitation. Although such an assessment is not required for scoping reviews, noting it would strengthen the readers’ understanding of the evidence base.

7. PLOS authors have the option to publish the peer review history of their article (what does this mean?). If published, this will include your full peer review and any attached files.). If published, this will include your full peer review and any attached files.

**Do you want your identity to be public for this peer review?** For information about this choice, including consent withdrawal, please see our Privacy Policy..

Reviewer #1: No

Reviewer #3: No

Figure Resubmissions:

---

## [Decision Letter · Decision Letter 2]

26 Mar 2026

Acceptance and commitment therapy (ACT)-based programs for reducing symptoms of depression, anxiety and/or stress applied to the educational context: an exploratory review.

PMEN-D-24-00459R2

**Dear Dr. Palacios,**

We are pleased to inform you that your manuscript 'Acceptance and commitment therapy (ACT)-based programs for reducing symptoms of depression, anxiety and/or stress applied to the educational context: an exploratory review.' has been provisionally accepted for publication in PLOS Mental Health.

Best regards,

Kizito Omona, PhD

Academic Editor

PLOS Mental Health

Reviewer Comments (if any, and for reference):

Reviewer's Responses to Questions

**Comments to the Author**

1. If the authors have adequately addressed your comments raised in a previous round of review and you feel that this manuscript is now acceptable for publication, you may indicate that here to bypass the “Comments to the Author” section, enter your conflict of interest statement in the “Confidential to Editor” section, and submit your "Accept" recommendation.

Reviewer #1: All comments have been addressed

Reviewer #3: All comments have been addressed

2. Does this manuscript meet PLOS Mental Health’s publication criteria? Is the manuscript technically sound, and do the data support the conclusions? The manuscript must describe methodologically and ethically rigorous research with conclusions that are appropriately drawn based on the data presented.? Is the manuscript technically sound, and do the data support the conclusions? The manuscript must describe methodologically and ethically rigorous research with conclusions that are appropriately drawn based on the data presented.

Reviewer #1: Yes

Reviewer #3: Yes

3. Has the statistical analysis been performed appropriately and rigorously?

Reviewer #1: N/A

Reviewer #3: N/A

4. Have the authors made all data underlying the findings in their manuscript fully available (please refer to the Data Availability Statement at the start of the manuscript PDF file)?

The PLOS Data policy requires authors to make all data underlying the findings described in their manuscript fully available without restriction, with rare exception. The data should be provided as part of the manuscript or its supporting information, or deposited to a public repository. For example, in addition to summary statistics, the data points behind means, medians and variance measures should be available. If there are restrictions on publicly sharing data—e.g. participant privacy or use of data from a third party—those must be specified.requires authors to make all data underlying the findings described in their manuscript fully available without restriction, with rare exception. The data should be provided as part of the manuscript or its supporting information, or deposited to a public repository. For example, in addition to summary statistics, the data points behind means, medians and variance measures should be available. If there are restrictions on publicly sharing data—e.g. participant privacy or use of data from a third party—those must be specified.

Reviewer #1: Yes

Reviewer #3: Yes

5. Is the manuscript presented in an intelligible fashion and written in standard English?

Reviewer #1: Yes

Reviewer #3: Yes

6. Review Comments to the Author

Reviewer #1: (No Response)

Reviewer #3: The authors have adequately addressed the comments raised in the previous round of review. I appreciate the authors’ efforts in responding to the concerns. I have no further comments.

7. PLOS authors have the option to publish the peer review history of their article (what does this mean?). If published, this will include your full peer review and any attached files.). If published, this will include your full peer review and any attached files.

**Do you want your identity to be public for this peer review?** For information about this choice, including consent withdrawal, please see our Privacy Policy..

Reviewer #1: No

Reviewer #3: No
